# Neighborhood Characteristics and the Mental Health of Caregivers Cohabiting with Care Recipients Diagnosed with Alzheimer’s Disease

**DOI:** 10.3390/ijerph18030913

**Published:** 2021-01-21

**Authors:** Dana M. Alhasan, Jana A. Hirsch, Chandra L. Jackson, Maggi C. Miller, Bo Cai, Matthew C. Lohman

**Affiliations:** 1Epidemiology Branch, National Institute of Environmental Health Sciences, National Institute of Health, Department of Health and Human Services, Research Triangle Park, NC 27709, USA; chandra.jackson@nih.gov; 2Urban Health Collaborative, Epidemiology and Biostatistics, Dornsife School of Public Health, Drexel University, Philadelphia, PA 19104, USA; jah474@drexel.edu; 3Intramural Program, National Institute on Minority Health and Health Disparities, National Institutes of Health, Department of Health and Human Services, Bethesda, MD 20814, USA; 4Department of Epidemiology and Biostatistics, Arnold School of Public Health, University of South Carolina, Columbia, SC 29208, USA; chandlmj@email.sc.edu (M.C.M.); BOCAI@mailbox.sc.edu (B.C.); LOHMANM@mailbox.sc.edu (M.C.L.)

**Keywords:** Alzheimer’s disease, caregivers, caregiving, dementia, mental health, poverty, poverty area, residence characteristics, rural health

## Abstract

While studies have documented the influence of caregiver and care recipient factors on caregiver health, it is important to address the potential impact of neighborhood contexts. This study estimated the cross-sectional associations between neighborhood characteristics and mental health among caregivers cohabiting with Alzheimer’s disease care recipients that were experiencing severe or non-severe neuropsychiatric symptoms (NPSs) (e.g., aggression/anxiety). We obtained data collected in 2010 on caregivers and care recipients (*n* = 212) from a subset of South Carolina’s Alzheimer’s Disease Registry. Neighborhood measures (within 1 mile of the residence) came from the American Community Survey and the Rural-Urban Commuting Area Code. We categorized the neighborhood median household income into tertiles, namely, “low” (<$31,000), “medium” ($31,000–40,758), and “high” (>$40,758), and rurality as “large urban,” “small urban,” and “rural.” We used negative binomial regression to estimate the prevalence ratios (PRs) and 95% confidence intervals (CIs) for caregiver mental health using neighborhood characteristics. The mean age was 58 ± 10.3 years, 85% were women, and 55% were non-Hispanic Black. Among the caregivers cohabiting with a recipient experiencing severe NPS, higher distress was experienced by caregivers living in low- (PR = 1.61 (95% CI = 1.26–2.04)) and medium- (PR = 1.45 (95% CI = 1.17–1.78)) vs. high-income neighborhoods after an adjustment. These results suggest that neighborhood characteristics may amplify other social stressors experienced by caregivers.

## 1. Introduction

An estimated 15 million people provide unpaid care for individuals experiencing Alzheimer’s disease (AD) and other forms of dementia in the United States (U.S.) [1]. AD caregivers, hereafter “caregivers,” bear substantial physical, mental, and financial burdens as they provide emotional support and assist with multiple activities of daily living (e.g., bathing), along with instrumental activities of daily living (e.g., paying bills) [1]. Given the demands associated with caregiving, caregivers often experience health problems [2,3,4]. Approximately 30–40% of caregivers suffer from depression compared to 5–17% of non-caregivers of similar ages [1]. Similarly, the prevalence of depression is higher among AD caregivers compared to other types of caregivers [5], potentially due to strong associations between neuropsychiatric symptoms (NPSs), such as delusions in people experiencing AD and depression in caregivers [6]. Most research on the influence of caregiving on caregiver health has examined individual-level factors that have an impact on poor caregiver mental health outcomes [7], which include older age, identifying as a woman, employment, spouse caregiving, taking care of a child, cohabitation with the AD care recipient [7], and severity of the NPSs of the recipient [1]. Yet, little research has focused on how the neighborhood context, such as where caregivers reside with care recipients, might influence the caregiver’s ability to provide care [8].

Extant literature on neighborhood environments and caregiver mental health is limited, although the research has suggested links between neighborhood characteristics and mental health conditions [9]. Studies have demonstrated the role between neighborhood disadvantage and other poor health outcomes [10] or depressive symptoms [9], suggesting that neighborhoods with greater exposure to adverse environmental stressors (e.g., crime) [11] may increase the risk of unfavorable mental health outcomes among caregivers [12]. Disadvantaged neighborhoods, which are often defined by the U.S. Census in terms of the composition of people living in the area, such as median household income [13], may also affect a caregiver’s vulnerability to stressors, thus potentially increasing their risk of experiencing depressive symptoms [14].

Residential instability, or the movement of people in and out of neighborhoods through time, has also been associated with greater depressive symptoms among adults [9]. For instance, a study in Cyprus found that lower social cohesion and fewer connections with neighbors was associated with greater caregiver burden among those taking care of someone with AD or related dementia [15]. Conversely, more connections with neighbors and greater social support may also influence aspects of caregiving. A systematic review concluded that a larger caregiver network and increased support were related to a lower burden among caregivers compared to caregivers with less social support [6].

Rural neighborhoods, which are generally characterized by limited access to healthcare, scarce resources, and geographic isolation, may also negatively impact caregivers’ mental health [16,17]. While no studies to our knowledge have examined the association between rurality and mental health among AD caregivers, recent research reported demographic differences between all types of caregivers living in rural compared with urban areas [18]. Caregivers in rural areas experienced lower employment, lower educational attainment, and lower income, which all relate to resource gaps and thus overall suggests a greater likelihood of caregiver burden in rural compared to urban areas [18]. Given this understudied topic [19,20], it is important to understand the impact of rurality on caregiving.

The anticipated increase in the aging population coupled with the growing prevalence of AD over the next forty years will lead to a greater reliance on caregivers; hence, it is vital to understand the risk factors that impact caregiver health [21]. In this study, we determined the associations between neighborhood characteristics (i.e., median household income, percent of residents who moved within the past year as a measure of residential instability, and rurality), and caregiver mental health, specifically depressive symptoms, caregiver burden, and caregiver distress, among caregivers cohabiting with their AD care recipient in 2010 in South Carolina (SC). We hypothesized that caregivers residing in neighborhoods with lower household incomes, a higher percentage of residents who recently moved, and more rural areas would have greater levels of depression, burden, and distress. We also hypothesized that the associations between neighborhood characteristics and mental health symptoms would be stronger among caregivers who cohabited with recipients experiencing severe NPSs compared to caregivers who cohabited with recipients without severe NPSs [22].

## 2. Methods

### 2.1. Data Sources

We obtained demographic data on both caregivers and cohabited care recipients from a subset of the SC Alzheimer’s Disease Registry [23]. The Registry is a comprehensive statewide roster of diagnosed cases of AD and other related dementias that is compiled from a variety of sources (e.g., inpatient hospitalizations, mental health records, and Medicaid) [24]. Data from the subsample were collected in 2010 by trained interviewers who asked caregivers by phone about their caregiving experiences and the care recipient’s behavioral disturbances. Caregivers were defined as a person who spends at least four hours per day and at least four days per week with the recipient. All recipients were diagnosed with AD between 2005 and 2010 based on the International Classification of Diseases (ICD)-9/10 Clinical Modifications codes (which may be made by a specialist or a general physician). Recipients with known vascular dementia or dementias caused by other medical conditions were excluded. All recipients were enrolled in a Medicaid waiver program (a federal and state program that helps with medical costs for some individuals with limited income and resources) and thus have comparable access to services. Finally, all recipients were eligible for a nursing home level of care, which included the option for the recipient and their caregivers to receive additional care services (e.g., transportation, home-delivered meals, and disposable medical supplies) and case management while still residing within the community. Further details regarding these services can be found elsewhere [25]. Most of the caregivers (96%) were family members of the recipient (e.g., children) and reported feeling a duty or responsibility to care for them, despite the recipients’ eligibility for long-term, institutionalized care. Further information regarding study details and eligibility criteria can be found elsewhere [23]. Each study participant provided informed consent for this study, which was deemed exempt by the Institutional Review Board at the University of South Carolina (ID = Pro00076582) and the National Institute of Environmental Health Sciences’ Institutional Review Board.

We used two secondary sources, which were both available online at the census tract level for neighborhood characteristics: the 2006–2010 American Community Survey (ACS) [26] and the 2010 U.S. Department of Agriculture Rural-Urban Commuting Area (RUCA) codes [27]. We obtained shapefiles and geographic features for SC data from the U.S. Census Topologically Integrated Geographic Encoding and Referencing (TIGER) Line Files [28].

### 2.2. Study Population: Caregiver Participants

The sample consisted of 224 caregivers who cohabited with care recipients. Twelve caregivers were excluded because we were unable to verify their residence (Appendix A). The remaining 212 caregivers were geocoded using ArcGIS Desktop Version 10.2.2 for Windows (Environmental Systems Research Institute, Redlands, CA, USA). One address was tied (i.e., had more than one assigned location with the same match score) and was re-matched using the “Interactive Rematch Dialogue” feature in ArcGIS. We compared the caregivers included in our study (*n* = 212) to the full caregiver sample (*n* = 224) and found no substantial differences (Appendix A).

### 2.3. Exposure Assessments: Neighborhood Characteristics

We defined a neighborhood as the 1-mile Euclidian (or radial) buffer distance around each caregiver’s geocoded address. Because individual residence-based buffers tend to overlap multiple census tracts, neighborhood characteristics were calculated as the weighted average of intersecting census tracts within the buffer. To account for variations in the relevant spatial scale, we conducted sensitivity analyses using 3-mile buffer distances. Caregiver neighborhood characteristics included both median household income and the percent of residents who moved within the past year. We categorized median household income per family into tertiles labeled “low” (<$31,000), “medium” ($31–40,758), and “high” (>$40,758). The ACS determined the extent of residential mobility by comparing data on the location of current residence and residence of one year ago.

We assigned caregiver rurality by the census tract in which the geocoded addresses resided. The RUCA measures rurality on a 10-point scale, ranging from metropolitan to rural. We divided the census tracts into three rurality categories: (1) “large urban” (a metropolitan area core; *n* = 105), (2) “small urban” (a metropolitan area with high commuting or a metropolitan area low commuting; *n* = 41), and (3) “rural” (a micropolitan area core, micropolitan area with high commuting, micropolitan area with low commuting, small town core, small town with high commuting, small town with low commuting, or rural areas; *n* = 66).

### 2.4. Outcome Assessments: Caregiver Mental Health

We considered three caregiver mental health outcomes: (1) depressive symptoms, (2) caregiver burden, and (3) caregiver distress (Appendix A). We measured depressive symptoms using the Center for Epidemiologic Studies Depression Scale–Revised (CESD-R), which is a validated self-report measure of depression [29] that has been recently validated among dementia caregivers [30]. The CESD-R is made up of ten statements regarding how one felt or behaved in the past week. Caregivers responded with 0 = rarely/none of the time, 1 = some of the time, 2 = occasionally, and 3 = most of the time. Summing each discrete response yielded a composite score. We measured the caregiver burden using the shortened Zarit Burden Interview (ZBI-4), which is a validated measure of caregiver burden [31]. The ZBI short version is made up of four items that caregivers ranked on a five-point scale: 0 = never, 1 = rarely, 2 = sometimes, 3 = quite frequently, and 4 = nearly always. We measured caregiver distress using the Neuropsychiatric Inventory Questionnaire (NPI-Q), which is a validated measure of both caregiver distress in relation to NPSs [32] and noncognitive symptoms among those with AD [33]. Caregivers reported the presence of 12 domains related to NPSs present among their care recipient (which we analyzed as a potential modifier): delusions, hallucinations, agitation/aggression, depression/dysphoria, anxiety, elation/euphoria, apathy/indifference, disinhibition, irritability/lability, motor disturbance, sleep and nighttime behavior disorders, and appetite/eating changes (Appendix A). For each of the 12 domains, caregivers rated both the severity of the symptoms on a three-point scale (1 = mild, 2 = moderate, and 3 = severe) and the frequency of the symptoms on a four-point scale (1 = occasionally, 2 = often, 3 = frequently, and 4 = very frequently). Multiplying the severity and frequency scores in each domain produced a domain score. The domain scores were summed across all twelve domains to ultimately yield a composite NPS score. For each present domain, caregivers assessed their level of distress by ranking it on a six-point scale: 0 = not distressing at all, 1 = minimal, 2 = mild, 3 = moderate, 4 = severe, and 5 = extreme or very severe. Summing each domain yielded a discrete composite score.

### 2.5. Potential Confounders

Caregiver demographic information obtained from the Registry subsample included current caregiver age, sex/gender (men or women), self-identified race/ethnicity (non-Hispanic (NH-)Black or other, including NH-White, Hispanic/Latinx, and Asian), employment (retired/unemployed, employed, or other), relationship to the care recipient (spouse, child, or other), and sandwich caregivers (yes or no). A sandwich caregiver refers to an individual who reported caretaking responsibilities for both individuals with AD and someone under 18 years old (e.g., a grandchild). Most studies identified person-level caregiver factors that influence mental health, which include older age, women, employed, spouse caregivers, sandwich caregivers, and cohabitation with the recipient [7]. We included these variables as potential confounders of the relationship between the neighborhood environment and the caregiver’s mental health.

### 2.6. Statistical Analyses

We computed descriptive statistics and presented categorical variables as numbers with percentages and continuous variables as means with standard deviations (SDs). To estimate the associations between neighborhood characteristics and caregiver mental health scores, we conducted a negative binomial regression that was stratified by the care recipient’s NPS severity status. Given the non-normal distribution of NPSs and recommendations to use non-parametric methods for NPSs [34], we used the median of the total NPS score (median = 19) to separate recipients into severe and non-severe statuses. We adjusted for the following confounders in the model: age, sex/gender, race/ethnicity, employment status, relationship with the care recipient, and sandwich caregiver status. We presented results obtained from these regression analyses as prevalence ratios (PRs), and we assessed diagnostics using the Pearson chi-square test of deviance. We did not include an offset parameter because the questionnaires asked about symptoms occurring during the same time frame. We set the significance level at 0.05 and completed all analyses using SAS software, version 9.3 for Windows (SAS Institute, Cary, NC, USA).

## 3. Results

### 3.1. Study Population Characteristics

Among the 212 caregivers cohabiting with their recipient, the mean age was 58.9 ± 10.3 years, the majority were women (85%), and over half of the caregivers were NH-Black (55.2%) (Table 1). Most caregivers lived in large urban neighborhoods (49.5%). The median scores were as follows: depression was 10 (S.D. = 6.36; range = 0–29), burden was 6 (S.D. = 3.95; range = 0–16), and distress was 10 (S.D. = 10.42; range = 0–45).

### 3.2. Neighborhood Characteristics and Caregiver Mental Health

Table 2, Table 3 and Table 4 present both the unadjusted and adjusted PRs modeling the average scores of depressive symptoms, burden, and caregiver distress, which were stratified by the care recipients’ NPS status, respectively. We estimated that for caregivers that cohabited with a recipient with severe NPSs, higher distress was experienced by caregivers living in low- (PR = 1.61 (95% CI = 1.26–2.04)) and medium- (PR = 1.45 (95% CI = 1.17–1.78)) vs. high-income neighborhoods after adjusting for the percent of residents who had moved within the past year, rurality, caregiver age, sex/gender, race/ethnicity, relationship to the recipient, employment, and sandwich caregiver status (Table 4). In contrast, the results suggested that caregivers of non-severe NPS recipients exhibited the opposite relationship between neighborhood income and depressive symptoms (low vs. high income: PR = 0.80 (95% CI = 0.55–1.17); medium vs. high income: PR = 0.77 (95% CI = 0.53–1.12)) (Table 2). Among caregivers for recipients with non-severe NPSs, we observed that those living in small urban areas had 37% (PR = 0.63 (95% CI = 0.31–1.27)) lower distress scores and those living in rural areas had 47% (PR = 0.53 (95% CI = 0.28–1.01)) lower distress scores vs. urban neighborhoods in adjusted models, although this failed to reach statistical significance (Table 4). We did not observe significant measures of association between neighborhood characteristics and caregiver burden or between residential instability and caregiver mental health outcomes. Sensitivity analyses that defined neighborhoods with a 3-mile buffer distance resulted in no significant differences (Appendix A). Additionally, modeling the average scores of caregiver mental health outcomes without considering the recipients’ severity status also resulted in no significant differences (Appendix A).

## 4. Discussion

Our study examined the impacts of neighborhood characteristics on caregiver mental health (i.e., levels of depression, burden, and distress) among those that cohabited with care recipients experiencing AD. The key study findings include evidence to support the hypothesis that caregivers that cohabited with recipients experiencing severe NPSs residing in low- vs. high-income neighborhoods were more likely to experience greater levels of poor mental health outcomes, in particular, distress. Notably, among caregivers that cohabited with non-severe NPSs, our results suggested that those living in low- vs. high-income neighborhoods experienced lower depressive symptoms and burden scores. Therefore, neighborhood characteristics may moderate caregiver outcomes among those living with a recipient with severe NPSs. Furthermore, the results suggested that caregivers residing in rural vs. urban areas experienced lower levels of mental health outcomes, which was inconsistent with our hypothesis. Lastly, we did not find an association between residential instability and caregiver mental health outcomes.

Few studies have sought to examine potential associations between neighborhood characteristics and caregiver mental health [8]. The current study extends the caregiver literature on mental health by illustrating differences in the association between neighborhood income and caregiver mental health related to the NPS severity of the care recipient. As previously noted, caregivers that cohabited with recipients with severe NPSs experienced greater distress when living in low- vs. high-income neighborhoods. Among caregivers of recipients with non-severe NPSs, we observed PRs suggesting that those living in lower-income neighborhoods experienced more positive mental health outcomes, although this was not statistically significant. The reasons for these differences are unclear. It may be that low-income neighborhoods are associated with greater caregiver distress because they lack available resources, such as respite care, specialty clinics, or caregiver support groups, to help caregivers manage the symptoms of their recipients. In this case, the association between low-income neighborhoods and poor mental health would be expected to be greatest among the caregivers of recipients with severe NPSs, as found in the present study. NPSs are some of the most commonly cited problematic behavioral symptoms caregivers manage while taking care of someone with AD [35]. The more severe NPSs of a recipient place a greater demand on the caregiver, which increases the caregiver’s risk of experiencing poor mental health [36,37]. Alternatively, previous research suggested that neighborhood disadvantage does not necessarily translate into poor caregiver mental health outcomes. For example, neighborhood disadvantage was associated with lower caregiver depression and was associated with more positive aspects of caregiving (e.g., feeling confident about one’s ability to provide care) [8], suggesting that neighborhood characteristics play a moderating role on the impact of individual-level risk factors on caregiver outcomes, particularly among caregivers living with a recipient with non-severe NPSs. There may be different expectations and inherent support networks concerning caregiving between low- and high-income neighborhoods; people living in low-income neighborhoods may be more likely to live in multigenerational homes where there is an inherent expectation that they will be called on to be caregivers. Thus, when there are no additional stressors (i.e., when the recipient does not have severe NPSs), caregivers in low-income neighborhoods benefit from this, leading to similar (or slightly better) mental health outcomes compared to caregivers living in high-income neighborhoods. On the other hand, when there are additional stressors on the caregiver (i.e., when the recipient has severe NPSs) that complicate the provision of care and go beyond their skills, they cannot rely on economic or other resources (e.g., home health, daycare) that would be enjoyed by caregivers in high-income neighborhoods. Thus, it is when the recipient has severe NPSs that differences in mental health emerge among caregivers living in low-income neighborhoods, who may be more taxed by the complicated caregiving demands. If there was more power to detect differences in our study, this idea might be reflected in an interaction between neighborhood income and NPSs. Nonetheless, most literature has demonstrated a relationship between disadvantaged neighborhoods and greater depressive symptoms [9,38], specifically among older adults [39,40,41], similar to our findings. Therefore, it is important to examine this relationship further among dementia caregivers in order to guide policies that better address the needs of caregivers (e.g., formal services like educational workshops).

Our findings suggest that residing in rural neighborhoods is associated with better caregiver mental health irrespective of care recipient NPS status. Particularly, we observed that caregivers living in rural vs. urban areas had lower distress scores on average, although this difference was not statistically significant. This is consistent with a recent meta-analysis that demonstrated that an urban compared to a rural residence was associated with greater depressive symptoms among those ≥60 years old [42], which is an age demographic that encompasses the majority of AD caregivers [1]. Other studies showed that dementia caregivers living in rural areas experienced less caregiving difficulties compared to those in urban areas, despite having an annual household income of <$25,000 or being unable to visit a doctor due to the financial cost [43]. While NPS severity among recipients is a known risk factor for depressive symptoms among caregivers [7], previous results suggest that neighborhood characteristics buffer the impact of NPSs on caregiver mental health [22]. The present findings suggest that rural areas may fulfill a similar function of providing a buffer for the effects of NPS, with one potential buffer being greater social support and stronger community ties. This hypothesis is supported by previous research which reported fairly high levels of social support (e.g., having available tangible material assistance, someone to discuss problems with, or positive regard and self-esteem from others) among dementia caregivers living in rural Alabama [44]. Previous research has also identified the availability of someone with whom to talk [44] and the ability to utilize places of worship (e.g., church) as sources for respite [19] as potential features of rural communities that may help caregivers better manage the challenges of dementia care, counteracting the limited availability of caregiving resources in these communities [44]. On the other hand, we did not observe significant differences in caregiver mental health outcomes due to residential instability, which is thought to influence depressive symptoms by hindering the formation of social cohesion and negatively impacting the support networks that are needed to protect individuals from worsening depressive symptoms [45]. This suggests that social cohesion and support are not the only explanation for rural/urban differences. Another potential explanation may be reporting bias. It is possible that caregivers reported experiencing fewer mental health problems in order to avoid judgment (e.g., social desirability bias). Additionally, mental health has been stigmatized among NH-Black adults [46], which is a racial group that makes up more than half of the caregivers; this stigmatization may influence how caregivers answer. In addition to stigmatization, general mental health may have been viewed differently by race/ethnicity. Typically, the literature reports lower depressive symptoms and poor mental health outcomes among NH-Black adults when compared to NH-White adults [47]. Nonetheless, the instrument used to measure depressive symptoms (CESD-R) had been validated among NH-Black older adults [48], NH-Black AD caregivers [49], and NH-Black caregivers in Missouri, where half of the sample consisted of those living in rural areas [50]. Despite the validation of CESD by race/ethnicity, the role of caregiving may also be viewed differently by racial/ethnic groups. Specifically, NH-Black caregivers may have different perceptions of caregiving compared to NH-White caregivers [51,52].

While we defined the neighborhood at a small scale, we assumed uniformity across the census tracts in order to use weighted administrative data. The small sample size may have limited our ability to detect effects. Future studies with larger sample sizes may also consider race/ethnicity as a potential modifier, considering that NH-Black individuals are more likely to live in low-income neighborhoods [53]. Additionally, the cross-sectional study design and the inability to assess changes in characteristics over time did not allow us to make causal conclusions about the associations between neighborhood characteristics and caregiver mental health. Longitudinal research will be helpful in identifying how changes in residence, neighborhood composition, or resources are related to changes in caregiver mental health. Longitudinal investigations of AD care recipients may be particularly informative for caregiver mental health when change is examined during critical transition periods (e.g., recipient behavioral changes or institutionalization). The cross-sectional design also did not account for neighborhood selection effects (i.e., selective sorting into neighborhoods). Although adjustment for individual-level data (e.g., caregiver employment) attempted to account for factors related to residential selection, residual confounding may exist. Potential selection bias regarding which caregivers chose to participate in the study may have limited the generalizability; specifically, caregivers with a recipient with greater NPS severity might have been less likely to respond due to having reduced time for an hour-long phone interview. However, we did observe more caregivers (*n* = 112) of recipients presenting severe NPSs compared to non-severe NPSs (*n* = 100). We are unable to assess the potential impact of this limitation, but it may constrain the representativeness of the sample population [23]. While the subsample used the ICD-9/10 to identify patients diagnosed with AD, it is possible that a general physician’s diagnosis compared to a specialist’s diagnosis may include patients with other dementias (e.g., vascular dementia), and we do not have data on who made the diagnosis to further explore this limitation. A strength of this study includes the use of validated and reliable questionnaires to capture depressive symptoms, caregiver burden, and distress score [29,32,54]. The availability of the Registry subsample data [23] allowed for the examination of neighborhood characteristics and caregiver mental health outcomes among a heterogenous, racially diverse population (30% lived in rural areas and 55% were NH-Black).

The current research adds knowledge to the literature regarding the role between lower-income neighborhoods and greater caregiver distress as related to the care recipients’ NPS severity status. Because we observed greater caregiver distress among those living in low- vs. high-income neighborhoods, future research can assess whether this relationship was specifically due to lower access to care, fewer opportunities for caregiver support groups, or other potential mechanisms. By examining the role of additional neighborhood characteristics, especially caregiver support groups, future research could focus on the explicit pathways between neighborhood environments and caregiver mental health. Additionally, because the role of low-income neighborhoods was greater among caregivers living with a recipient with severe NPSs, the mitigation of NPSs may help improve caregiver health. Similarly, perceived caregiver distress due to NPSs may be another potential mediator for the observed relationships between neighborhood characteristics and caregiver mental health. Thus, interventions can be specifically tailored to these caregivers who may be at higher risk for distress.

## 5. Conclusions

Our study adds to an important area of research, considering the anticipated growing burden on caregivers and the lack of effective treatments for AD. Overall, we observed that caregivers that cohabited with AD care recipients presenting severe NPSs and living in low-income neighborhoods experienced greater caregiver distress. Caregiver mental health was greatly associated with care recipients’ NPSs and disease progression; in cases where NPSs persisted, they increased the burden on the caregiver indirectly by increasing the risk for institutionalization, comorbidities, and mortality [54]. These results suggest that neighborhood characteristics may serve to amplify other social stressors experienced by caregivers. Therefore, it is important for policies and interventions to address caregivers’ needs by targeting communities. Additionally, broader neighborhood policies, such as providing opportunities for physical activity, increasing access to healthy food, and promoting community cohesion, can improve overall health and well-being in the community. Furthermore, potential clinical applications may include the mitigation of NPSs, which not only relieves caregivers but also those with AD. This study supports an approach to identifying neighborhood environmental characteristics that influence caregiver mental health (e.g., stressors, like unemployment, in low-income neighborhoods) in order to offer community-level interventions to alleviate caregiver depression, burden, and distress.

## Figures and Tables

**Table 1 ijerph-18-00913-t001:** Descriptive characteristics of cohabited caregivers’ neighborhood and demographic characteristics, 2010 (*n* = 212).

Neighborhood Characteristics	*N* (%)	Caregiver Depression		Caregiver Burden		Caregiver Distress	
		<Median	>Median	<Median	>Median	<Median	>Median
Percent moved 1 year ago, mean (S.D.)	3.9 ± 1.9	3.7 ± 1.9	3.9 ± 1.8	3.8 ± 1.7	3.9 ± 1.9	3.9 ± 1.9	3.7 ± 1.7
Rurality ^a^							
Large urban	105 (49.5)	48 (45.7)	57 (53.2)	44 (44.0)	61 (54.4)	44 (42.3)	61 (56.4)
Small urban	41 (19.3)	23 (21.9)	18 (16.8)	18 (18.0)	23 (20.5)	21 (20.1)	20 (18.5)
Rural	66 (31.1)	34 (32.3)	32 (29.9)	38 (38.0)	28 (25.0)	39 (37.5)	27 (25.0)
Median household income							
High (>$40,758)	54 (25.4)	36 (34.2)	34 (31.7)	25 (25.0)	46 (41.1)	34 (32.6)	37 (34.2)
Medium ($31–40,758)	96 (45.2)	34 (32.3)	37 (34.5)	38 (38.0)	33 (29.4)	37 (35.5)	34 (31.4)
Low (<$31,000)	62 (29.2)	35 (33.3)	36 (33.6)	37 (37.0)	33 (29.4)	33 (31.7)	37 (34.2)
**Caregiver Demographics**	***N*** **(%)**	**<Median**	**>Median**	**<Median**	**>Median**	**<Median**	**>Median**
Age, mean (S.D.)	58.9 ± 10.3	58.9 ± 10.9	58.9 ± 9.7	58.8 ± 10.9	59.1 ± 9.8	59.3 ± 10.6	58.6 ± 10.1
Sex/gender							
Male	31 (14.6)	18 (17.1)	13 (12.1)	21 (21.0)	10 (8.9)	23 (22.1)	8 (7.4)
Female	181 (85.3)	87 (82.8)	94 (87.8)	79 (79.0)	102 (91.1)	81 (77.8)	100 (92.5)
Race/ethnicity							
Non-Hispanic Black	117 (55.1)	66 (62.8)	51 (47.6)	67 (67.0)	50 (44.6)	64 (61.5)	53 (49.1)
Other ^b^	95 (44.8)	39 (37.1)	56 (52.3)	33 (33.0)	62 (55.3)	40 (38.4)	55 (50.9)
Employment ^c^							
Employed	70 (33.0)	35 (33.3)	35 (16.5)	56 (56.0)	40 (35.7)	40 (38.4)	30 (27.7)
Retired/unemployed	113 (53.3)	56 (53.3)	57 (53.2)	30 (30.0)	57 (50.8)	52 (50.0)	61 (56.4)
Other	29 (13.6)	14 (13.3)	15 (14.1)	14 (14.0)	15 (13.4)	12 (11.5)	17 (15.7)
Relationship to the care recipient							
Spouse	35 (16.5)	12 (11.4)	23 (21.5)	16 (16.0)	19 (16.9)	15 (14.4)	20 (18.5)
Child	144 (67.9)	72 (68.5)	72 (67.2)	67 (67.0)	77 (68.7)	72 (69.2)	72 (66.6)
Other ^d^	33 (15.5)	21 (20.0)	12 (11.2)	17 (17.0)	16 (14.2)	17 (16.3)	16 (14.8)
Sandwich caregivers ^e^							
Yes	72 (33.9)	40 (38.1)	32 (29.9)	40 (40.0)	32 (28.5)	36 (34.6)	36 (33.3)
No	140 (66.1)	65 (61.9)	75 (70.1)	60 (60.0)	80 (71.4)	68 (65.3)	72 (66.6)
**Care Recipient Demographics**	***N*** **(%)**	**<Median**	**>Median**	**<Median**	**>Median**	**<Median**	**>Median**
Neuropsychiatric symptom severity (NPS) mean (S.D.)	26.3 ± 22.3	18.7 ± 15.6	33.7 ± 25.3	19.7 ± 20.9	32.2 ± 21.9	10.6 ± 8.6	41.4 ± 21.1
Severe NPS (>median)	112 (52.8)	43 (40.9)	69 (64.4)	38 (38.0)	74 (66.1)	17 (16.3)	95 (87.9)
Non-severe NPS (<median)	100 (47.1)	62 (59.1)	38 (35.5)	62 (62.0)	38 (33.9)	87 (83.6)	13 (12.1)
Age, mean (S.D.)	82.4 ± 8.7	82.6 ± 8.4	82.2 ± 8.96	81.9 ± 9.0	82.8 ± 8.5	82.4 ± 9.1	82.4 ± 8.4
Sex/gender							
Male	58 (27.3)	27 (25.7)	31 (28.9)	30 (30.0)	28 (25.0)	27 (25.9)	31 (28.7)
Female	154 (72.6)	78 (74.2)	76 (71.0)	70 (70.0)	84 (75.0)	77 (74.0)	77 (71.3)

^a^ Rurality was measured based on the RUCA (Rural-Urban Commuting Area codes). A “large urban” area was defined as a metropolitan area; a “small urban” area was defined as a metropolitan area with high commuting or a metropolitan area with low commuting; a “rural” area was defined as a micropolitan area core, micropolitan area with high commuting, micropolitan area with low commuting, small town core, small town with high commuting, small town with low commuting, or rural areas. ^b^ “Other” race/ethnicity included non-Hispanic White (*n* = 93), Hispanic/Latinx (*n* = 1), and Asian (*n* = 1). ^c^ Retired and unemployed included fully retired (*n* = 64), unemployed (*n* = 31), and homemaker (*n* = 18), and employed included employed full time (*n* = 40), employed part time (*n* = 27), and retired but working part time (*n* = 3). ^d^ Other relationship to the care recipient included daughter-in-law (*n* = 6), sister (*n* = 5), brother (*n* = 3), grandchild (*n* = 9), niece or nephew (*n* = 2), and other (*n* = 8). ^e^ Sandwich caregivers were defined as those who reported also taking care of someone under 18 years old (e.g., a grandchild). The median scores of caregiver depression symptoms, burden, and distress were 10, 6, and 10.

**Table 2 ijerph-18-00913-t002:** Unadjusted and adjusted prevalence ratios (PRs) of caregiver depressive symptoms by neighborhood characteristics, 2010.

Variable	Severe Neuropsychiatric Symptoms (*n* = 112) ^c^		Non-Severe Neuropsychiatric Symptoms (*n* = 100) ^c^	
	Unadjusted PR (95% CI)	Adjusted PR ^d^ (95% CI)	Unadjusted PR (95% CI)	Adjusted PR ^d^ (95% CI)
Rurality ^a^				
Rural	1.06 (0.83–1.35)	0.98 (0.75–1.27)	0.89 (0.65–1.22)	0.98 (0.69–1.39)
Small urban	0.88 (0.66–1.19)	0.96 (0.71–1.28)	0.89 (0.62–1.28)	1.02 (0.68–1.51)
Large urban	1.00 ^e^	1.00 ^e^	1.00 ^e^	1.00 ^e^
Percent moved 1 year ago	1.02 (0.96–1.09)	1.04 (0.95–1.12)	0.99 (0.93–1.07)	0.98 (0.89–1.09)
Median household income ^b^				
Low	1.18 (0.91–1.53)	1.33 (1.00–1.78)	0.82 (0.59–1.14)	0.80 (0.55–1.17)
Medium	1.19 (0.93–1.54)	1.25 (0.97–1.61)	0.75 (0.53–1.05)	0.77 (0.53–1.12)
High	1.00 ^e^	1.00 ^e^	1.00 ^e^	1.00 ^e^

^a^ Rurality was measured based on the RUCA (Rural-Urban Commuting Area codes). A “large urban” area was defined as a metropolitan area; a “small urban” area was defined as a metropolitan area with high commuting or a metropolitan area with low commuting; a “rural” area was defined as a micropolitan area core, micropolitan area with high commuting, micropolitan area with low commuting, small town core, small town with high commuting, small town with low commuting, or rural areas. ^b^ Median household income categories were based on tertiles, where “low” was <$31,000, “medium” was $31–40,758, and “high” was >$40,758. ^c^ Severe neuropsychiatric symptoms among those with AD were defined as >median. ^d^ Model was adjusted for caregiver age, sex/gender, race/ethnicity, employment, relationship to the care recipient, and sandwich caregiver status. ^e^ Reference category.

**Table 3 ijerph-18-00913-t003:** Unadjusted and adjusted prevalence ratios of caregiver burden by neighborhood characteristics, 2010.

Variable	Severe Neuropsychiatric Symptoms (*n* = 112) ^c^		Non-Severe Neuropsychiatric Symptoms (*n* = 100) ^c^	
	Unadjusted PR (95% CI)	Adjusted PR ^d^ (95% CI)	Unadjusted PR (95% CI)	Adjusted PR ^d^ (95% CI)
Rurality ^a^				
Rural	0.87 (0.67–1.12)	0.87 (0.65–1.18)	0.89 (0.30–1.32)	0.99 (0.66–1.51)
Small urban	0.93 (0.69–1.25)	1.02 (0.74–1.40)	0.86 (0.56–1.37)	0.94 (0.59–1.50)
Large urban	1.00 ^e^	1.00 ^e^	1.00 ^e^	1.00 ^e^
Percent moved 1 year ago	1.02 (0.96–1.08)	1.02 (0.94–1.12)	1.00 (0.92–1.10)	0.95 (0.83–1.07)
Median household income ^b^				
Low	0.91 (0.70–1.19)	1.05 (0.77–1.43)	0.88 (0.59–1.33)	0.90 (0.57–1.44)
Medium	0.94 (0.72–1.21)	1.01 (0.77–1.33)	0.65 (0.42–0.99)	0.74 (0.47–1.17)
High	1.00 ^e^	1.00 ^e^	1.00 ^e^	1.00 ^e^

^a^ Rurality was measured based on the RUCA (Rural-Urban Commuting Area codes). A “large urban” area was defined as a metropolitan area; a “small urban” area was defined as a metropolitan area with high commuting or a metropolitan area with low commuting; a “rural area” was defined as a micropolitan area core, micropolitan area with high commuting, micropolitan area with low commuting, small town core, small town with high commuting, small town with low commuting, or rural areas. ^b^ Median household income categories were based on tertiles, where “low” was <$31,000, “medium” was $31–40,758, and “high” was >$40,758. ^c^ Severe neuropsychiatric symptoms among those with AD were defined as >median. ^d^ Model was adjusted for caregiver age, sex/gender, race/ethnicity, employment, relationship to the care recipient, and sandwich caregiver status. ^e^ Reference category.

**Table 4 ijerph-18-00913-t004:** Unadjusted and adjusted prevalence ratios of caregiver distress by neighborhood characteristics, 2010.

Variable	Severe Neuropsychiatric Symptoms (*n* = 112) ^c^		Non-Severe Neuropsychiatric Symptoms (*n* = 100) ^c^	
	Unadjusted PR (95% CI)	Adjusted PR ^d^ (95% CI)	Unadjusted PR (95% CI)	Adjusted PR ^d^ (95% CI)
Rurality ^a^				
Rural	0.93 (0.75–1.15)	0.76 (0.61–0.95)	0.66 (0.39–1.13)	0.53 (0.28–1.01)
Small urban	0.95 (0.74–1.23)	0.90 (0.70–1.15)	0.66 (0.36–1.22)	0.63 (0.31–1.27)
Large urban	1.00 ^e^	1.00 ^e^	1.00 ^e^	1.00 ^e^
Percent moved 1 year ago	1.00 (0.95–1.06)	0.99 (0.92–1.06)	1.02 (0.90–1.16)	0.93 (0.78–1.12)
Median household income ^b^				
Low	**1.28 (1.02–1.59)**	**1.61 (1.26–2.04)**	1.13 (0.64–2.00)	1.50 (0.73–3.08)
Medium	**1.25 (1.00–1.55)**	**1.45 (1.17–1.78)**	0.86 (0.48–1.56)	1.20 (0.63–2.28)
High	1.00 ^e^	1.00 ^e^	1.00 ^e^	1.00 ^e^

^a^ Rurality was measured based on the RUCA (Rural-Urban Commuting Area codes). A “large urban” area was defined as a metropolitan area; a “small urban” area was defined as a metropolitan area with high commuting or a metropolitan area with low commuting; a “rural” area was defined as a micropolitan area core, micropolitan area with high commuting, micropolitan area with low commuting, small town core, small town with high commuting, small town with low commuting, or rural areas. ^b^ Median household income categories were based on tertiles, where “low” was <$31,000, “medium” was $31–40,758, and “high” was >$40,758. ^c^ Severe neuropsychiatric symptoms among those with AD were defined as >median. ^d^ Model was adjusted for caregiver age, sex/gender, race/ethnicity, employment, relationship to the care recipient, and sandwich caregiver status. ^e^ Reference category. Bolded estimates indicate statistical significance.

## Data Availability

Restrictions apply to the availability of these data. Data was obtained from the South Carolina Revenue and Fiscal Affairs Office, in partnership with the Office for the Study of Aging at the University of South Carolina.

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
