# Peer review of "Neighborhood Characteristics and the Mental Health of Caregivers Cohabiting with Care Recipients Diagnosed with Alzheimer’s Disease"

_ijerph, 2021, doi:10.3390/ijerph18030913_

Round 1

Reviewer 1 Report

This is an interesting study that evaluated cross-sectional associations between neighborhood characteristics and mental  health among caregivers cohabiting with Alzheimer’s Disease care recipients experiencing neuropsychiatric symptoms.

Major limitations of the study are represented by the small sample size and potential selection bias of caregivers included in the analysis; whereas strength points are represented by the use of validated and reliable questionnaires and the analysis of an heterogeneous population. Nonetheless, the Authors have already provided an exhaustive list of limitations and strengths of the study in the discussion paragraph

In general, the manuscript is well written; please revise the English language and check throughout the text for spelling errors. The Methods section is clear and well described.

Tables are detailed and helpful for the reader. Please check the title “Supplemental Figure 1. Flowchart of Caregiver Case Selection, 20” as it seems that “20” is incomplete; also, the writings in some diagrams are incomplete and/or not visible.

Please add square brackets in the text for references number, according to the guidelines of the Journal.  

Finally, I would include further discussion in the conclusion paragraph on future directions and potential clinical applications of the results.

Reviewer 2 Report

Neighborhood characteristics and the mental health of caregivers cohabiting with care recipients diagnosed with Alzheimer’s Disease

This manuscript reported that caregivers cohabiting with a patient with Alzheimer’s disease experiencing severe neuropsychiatric symptoms, higher distress was experienced by caregivers living in low and medium income compared to high income neighborhoods. This findings are very important for the caregivers of AD patient.

However, there are several concerns in this manuscript. Followings are my comments to authors.

Major concerns

  1. Although this manuscript used the data of an AD registry, it might be difficult for the readers to understand who made a diagnosis of AD, how to diagnose AD, and the severity of AD. Authors should add who made a diagnosis of AD (e.g. neurologist, psychiatrist, general physician), the diagnostic criteria used in AD registry, and the severity of AD such as the score of MMSE and ADAS-CoG. The clinical diagnosis of AD is usually difficult for general physician except for neurologist and psychiatrist who is professional for dementia. The mere presence of cognitive impairment is not sufficient for the diagnosis of AD. For example, AD specific past history such as more severely impaired memory function compared to other cognitive functions at least in the early stage is important. Brain MRI findings (such as atrophy of hippocampus and temporal lobe) and the findings of PET or SPECT revealing markedly reduced blood flow in precuneus might also be helpful for the diagnosis of AD. In my opinion, “AD patients diagnosed by general physician (not the expertise neurologist or psychiatrist)” include not only the AD patients but also include vascular dementia, dementia with Lewy body (DLB). Although it is not uncommon that AD patients also have comorbid vascular dementia or DLB, detailed differentiation of dementia should be performed as far as possible.
  2. AD patients (care recipient) are dichotomized to “severe NPS group” and “ non-severe NPS group” based on the median total of NPS score. I do not know whether it is statistically appropriate or not to dichotomize AD patients based on median score of NPS (probably) simply because NPS showed non-normal distribution. At least, distribution of NPS score should be provided as figures. The authors should discuss the plausible reason for using median score rather than more statistically relevant method.
  3. The detailed prevalence and the severity of neuropsychiatric symptoms such as delusion, agitation, disinhibition, irritability and night-time behavior disorders should be added, because these symptoms might significantly influence caregiver’s burden. Furthermore, motor disturbance are not the symptoms of AD. Was night time behavior disorders are completely discriminated from REM sleep behavior disorders (RBD) which is common in DLB rather than AD? Again, it is very important who made a diagnosis of AD in this registry.
  4. The reason why caregivers of recipients with non-severe NPS living in lower income neighborhoods experienced more positive mental health should be discussed in more detail, because this findings are very difficult to understand.
  5. The availability of care insurance system might influence the caregiver’s burden. Brief explanations for care insurance system in USA is necessary.

Minor concerns

  1. All tables are difficult to understand and are should be revised more clearly. Please adjust the position of each values (particularly table 1) . The positions of the title of parameter (caharacteristics, demographics, variable) and the corresponding values are not aligned.
  2. In results section 3.2, each PR values should be represented with corresponding table number.
  3. Although “low vs. high-income: PR= 0.88 [95% CI= 0.55-1.17]” are found in line 215, “PR=0.80 (0.55-1.17)” are found in table 2.

Round 2

Reviewer 2 Report

This manuscript is revised correctly and might be acceptable for publication in its current form.